# Viroinformatics-Based Analysis of SARS-CoV-2 Core Proteins for Potential Therapeutic Targets

**DOI:** 10.3390/antib10010003

**Published:** 2021-01-11

**Authors:** Lokesh Agrawal, Thanasis Poullikkas, Scott Eisenhower, Carlo Monsanto, Ranjith Kumar Bakku, Min-Hua Chen, Rajkumar Singh Kalra

**Affiliations:** 1Universidad Integral del Caribe y América Latina, Kaminda Cas Grandi #79, Willemstad, Curacao; 2Graduate School of Comprehensive Human Sciences, University of Tsukuba, 1-1-1 Tennodai, Tsukuba 305-8577, Japan; 3Human Biology, School of Integrative and Global Majors, University of Tsukuba, 1-1-1 Tennodai, Tsukuba 305-0006, Japan; s1830525@s.tsukuba.ac.jp (T.P.); tsukubascott0@gmail.com (S.E.); 4Department of Experimental Pathology, Faculty of Medicine, University of Tsukuba, 2-1-1 Tennodai, Tsukuba 305-8576, Japan; 5Department of Infection Biology, Faculty of Medicine, University of Tsukuba, 1-1-1 Tennodai, Tsukuba 305-8575, Japan; 6Research Workgroup, Ronin Institute, 127 Haddon Place, Montclair, NJ 07043-2314, USA; carlo.monsanto@iolee.life; 7Department of Computer Science, Faculty of Engineering Information and Systems, University of Tsukuba, 1-1-1 Tennodai, Tsukuba 305-8572, Japan; ranjithkumar.bakku@gmail.com; 8Tsukuba Life Science Innovation Program (TLSI), University of Tsukuba, 1-1-1 Tennodai, Tsukuba 305-8572, Japan; 9Department of Biomedical Engineering, Chung Yuan Christian University, 200, Chung Pei Road, Taoyuan City 32023, Taiwan; chen.minhua@cycu.edu.tw; 10AIST-INDIA DAILAB, National Institute of Advanced Industrial Science and Technology (AIST), Tsukuba 305-8565, Japan

**Keywords:** COVID-19/SARS-CoV-2/coronavirus 2, lower respiratory tract diseases, Spike protein, RdRp, hypoxia

## Abstract

SARS-CoV-2 (severe acute respiratory syndrome coronavirus 2) is a novel coronavirus for which no known effective antiviral drugs are available. In the present study, to accelerate the discovery of potential drug candidates, bioinformatics-based in silico drug discovery approaches are utilized. We performed multiple sequence alignments of the Spike (S) protein with 75 sequences of different viruses from the Orthocoronavirinae subfamily. This provided us with insights into the evolutionarily conserved domains that can be targeted using drugs or specific antibodies. Further, we analyzed the mechanism of SARS-CoV-2 core proteins, i.e., S and RdRp (RNA-dependent RNA polymerase), to elucidate how the virus infection can utilize hemoglobin to decrease the blood oxygen level. Moreover, after a comprehensive literature survey, more than 60 antiviral drugs were chosen. The candidate drugs were then ranked based on their potential to interact with the Spike and RdRp proteins of SARS-CoV-2. The present multidimensional study further advances our understanding of the novel viral molecular targets and potential of computational approaches for therapeutic assessments. The present study can be a steppingstone in the selection of potential drug candidates to be used either as a treatment or as a reference point when designing a new drug/antibody/inhibitory peptide/vaccine against SARS-CoV-2.

## 1. Introduction

In the past two decades, the world has faced several infectious disease outbreaks, such as influenza A (H1N1), SARS, MERS, Ebola, and Zika virus. These have had an enormous global impact in terms of healthcare, causing significant economic distress for nations. Most recently, there has been the global outbreak of the novel coronavirus, or severe acute respiratory syndrome coronavirus 2 (SARS-CoV-2), which is causing coronavirus disease 2019 (COVID-19). The World Health Organization (WHO) reported that the present worldwide pandemic originated from the Chinese city of Wuhan [1,2,3]. This newly discovered coronavirus strain—SARS-CoV-2—has a high transmission rate, causing severe disease in the lower portion of the respiratory tract. Through epidemiological data, scientists have been able to map the route of person-to-person transmission in contracting SARS-CoV-2, accounting for the rapid spread, causing the worldwide public health challenge [2]. COVID-19 patients may develop pneumonia-like symptoms which can deteriorate rapidly into respiratory failure. The elderly and, in general, people with compromised immune function have a higher disease susceptibility and mortality rate. According to the latest report from the WHO, 43.7 million people have suffered from COVID-19 and the death-toll has grown beyond more than one million worldwide. Of all the countries that have contributed to this total number of infections, the USA and Europe account for 8.7 million and 5.9 million cases, respectively. Together, they comprise more than one-third of known cases worldwide. Despite cautious optimism at the recent decline in cases, there is still fear of subsequent waves of infections after the relaxation of containment procedures in various cities/countries. Therefore, to protect human lives, discovering an effective target and inhibitor of SARS-CoV-2 is of utmost importance [4,5,6,7].

During the early stages of the SARS-CoV-2 pandemic, researchers and clinicians scrambled to find an effective treatment for those suffering from the virus. Unfortunately, in the light of elusive mechanisms that contribute to its pathogenesis, no efficient therapeutics regime can be adopted to combat it. To find an effective cure for COVID-19, we essentially need to understand its structure and infection mechanism. Structurally, SARS-CoV-2 is a single stranded (ss) RNA virus with a genome that consists of ~30,000 nucleotides. This encodes four structural proteins—nucleocapsid (N) protein, membrane (M) protein, Spike (S) protein, envelop (E) protein—and ~16 non-structural proteins (nsp) [8,9,10]. The Spike protein is integrated across the surface of the virus; it mediates attachment of the virus to the host cell surface receptors and the fusion between the viral and host cell membranes to facilitate entry into the host cell [11,12]. In addition, it is subjected to proteolytic cleavage by host proteases (i.e., trypsin and furin) in two sites at the polybasic furin cleavage site (PCS) located at the boundary between the S1 and S2 subunits (S1/S2 site). Binding of ACE2 with Spike S1 (AA: 15–685) proteins allows the virus to adhere to lung epithelial cells, and through a tube composed of Spike S2 proteins (AA: 730–1273), it injects the genetic material inside the host cell. Once the virus reaches the budding stage, the S2 domain is cleaved in order to release the fusion peptide [13].

Due to the rapid changes in the genome of SARS-CoV-2, its structural proteins are continuously evolving. Therefore, to find out the therapeutically important conservative protein motifs of the Spike protein, we performed multiple sequence alignments between the Spike protein sequences of SARS-CoV-2 and 75 sequences from different virus species that belong to the Orthocoronavirinae subfamily. Furthermore, clinical trial reports suggest that patients suffering from COVID-19 have a low oxygen concentration in their blood not caused solely by respiratory distress [14,15,16]. This phenomenon led us to hypothesize a possible binding event between SARS-CoV-2’s Spike protein and the hemoglobin, specifically heme group interaction [17,18]. Given the current need to reduce cost, time, and effort in the drug development process, researchers are exploring the repurposing of approved existing drug candidates to test for reducing disease symptoms in COVID-19 patients. The repurposing of FDA-approved/preclinical trial drugs to target multiple proteins of SARS-CoV-2 provides a ray of hope for the survival of COVID-19 patients in the present global health crisis. In this work, we thoroughly investigated the structure and conserved motifs of the SARS-CoV-2 core proteins. We docked these therapeutically important motifs with a series of repurposed drugs, including antiviral, anti-bacterial, and anti-parasitic drugs, as well as flavonoids and vitamins, shown to be effective against other coronaviruses and similar pathogens. The present report provides insights into the structural motifs of the SARS-CoV-2 core proteins and assessed therapeutic potency of existing drug candidates against SARS-CoV-2.

## 2. Materials and Methods

### 2.1. Investigation of Conservative Motifs of SARS-CoV-2 Spike Protein and Phylogenetic Analysis

To investigate the SARS-CoV-2 Spike protein’s conservative motifs and submit it to phylogenetic analysis, we used the Blast Molecular Evolutionary Genetics Analysis (MEGA 10.0.5) tool. In search of unique conservative regions in the Spike protein throughout its evolution, we performed alignment of the amino acid sequence of the Spike protein with the BLASTP database. We then selected 75 subject sequences of different viruses from the Orthocoronavirinae subfamily, which had the highest BLASTP score. The maximum number of possible alignments in one run determined the number of sequences. Later, we ran the alignment of these 75 sequences from the database using the MEGA-X software’s MUSCLE program [19]. Further, a phylogenetic tree was generated by using the maximum likelihood method and the JTT matrix-based model [20]. We then selected the tree with the highest log-likelihood score for the top 17 virus strains that are phylogenetically closely related to the novel coronavirus. The percentage of trees in which the associated taxa clustered together is shown next to the branches that were obtained. This analysis involved 17 amino acid sequences. There were a total of 1728 positions in the final dataset. The phylogeny was tested using the resampling bootstrap methodology with a bootstrap value of 100 [19].

The five selected species that underwent detailed alignment evaluations are as follows: Chain A Spike Glycoprotein Porcine epidemic diarrhea virus (PDB ID: 6VV5_A), Chain A S protein Middle East respiratory syndrome MERS (PDB ID: 6PZ8_A), Chain A Feline Infectious Peritonitis Virus Spike Protein FIP-CoV (PDB ID: 6JX7_A), Chain A Spike Glycoprotein SARS-CoV BJ01 (PDB ID: 5X58_A), Chain A Spike glycoprotein Human-CoV NL63 (PDB ID: 5SZS_A).

### 2.2. In Silico Protein-Protein Interaction

We performed the protein–protein docking using ClusPro (v2.0) server [21]. We submitted macromolecules in .pdb format to the ClusPro (v2.0) server and downloaded the most favorable interaction model for further analysis in Discovery Studio Visualizer (DSV) (v2019 client; Dassault Systèmes BIOVIA). We docked the SARS-CoV-2 Spike protein with both the tetramer and monomer molecules of hemoglobin (1A3N.pdb) and the heme group (extracted from 1A3N.pdb), in ClusPro server and AutoDockTools (v1.5.6) software, respectively. LigPlot + v.2.2 software was used to generate a 2D plot of interrace amino acids at the Hemoglobin-Spike interaction pocket [22].

### 2.3. Finding the Most Interacting Motifs of Spike Protein with Antiviral Drugs and Heat-Map Representation of Motifs

We manually investigated the binding pocket of Spike protein and RdRp with the docked drug ligands. We selected antiviral FDA approved drugs, flavonoids, and antibiotics, which are readily available on the market. In addition to that, these drugs are under investigation for their potency in the treatment of COVID-19. We selected the drugs from the PubChem database (https://pubchem.ncbi.nlm.nih.gov/), which is a national library of medicines managed by National Institute of Health, USA. We divided the whole protein sequence of Spike and RdRp proteins in, respectively, 12 and 10 equally spanned motifs. Based on the number of interactions with the docked drug ligand database, we ranked each motif and further normalized the number of interactions in each motif to compare their potential as a target for the therapeutic drugs and their significance for future drug development against SARS-CoV-2. In continuation, based on the normalization of the total number of interactions for each motif, heat-maps were produced using DSV. Similarly, we investigated the frequency and location of specific amino acids of Spike and RdRp in each motif, which are important for the interaction with various drug ligands.

### 2.4. Docking of Antivirals, Antibiotics, Antiparasitics, Flavonoids and Vitamins with Spike and RdRp Protein of SARS-CoV-2

Three-dimensional models of SARS-CoV-2 Spike (QHD43416.pdb) and RdRp (QHD43415_11.pdb) proteins were downloaded from Zhang Lab I-TESSAR online library in .pdb format. We downloaded 3D structures of drugs and small molecules for docking from PubChem in .pdb format [23]. Literature suggested, there is no significant change in the structure of these drug molecules after getting activated inside the liver, which makes them a suitable candidate for the docking study. Docking was performed with the AutoDockTool (v1.5.6) software [24]. There was no change observed in these drug molecules’ structure after being activated inside the liver, which makes them a suitable candidate for the docking study. Docking was performed with the AutoDockTool (v1.5.6) software [24]. All models were converted to “pdbqt” format after deleting water molecules and computing the Gasteiger algorithms for the macromolecules, whereas, in ligands, we choose torsion angles ≤ 9. For developing the grid box for SARS-CoV-2 Spike protein, we set the following metrics: X-dimension: 104, Y-dimension: 116, Z-dimension: 126, Spacing: 1.000, Offset X: 2.917, Offset Y: 1.450 and Offset Z: 0.194 and grid was set using Autogrid4. Docking was performed with Autodock4; for each docking, we set our macromolecules’ Rigid filename and docked with Genetic algorithm and output set to Lamarckian GA (4.2). After the docking simulation, 10 docking models were analyzed and ranked by their energy binding ability (∆G). H-bonds were also built in the docking models, and the best model of each docking was saved as a “.pdbqt” format file. Docking models were further analyzed for the discovery of a binding pocket in Discovery Studio Visualizer (DSV) (v2019 client; Dassault Systèmes BIOVIA) software. Interactions were built by selecting all favorable interaction types intermolecularly. Furthermore, 2D diagrams were visualized through DSV.

## 3. Results

### 3.1. Phylogenetic Analysis of SARS-CoV-2 Spike Protein and Retrieval of Therapeutically Important Conservative Motifs of Evolutionary Significance

We ran the sequence alignment of 75 subject sequences from the database using MEGA-X software and developed a phylogenetic tree through BLASTP. The subject sequences suggested by BLASTP are related to the Spike (S) proteins from MERS, murine hepatitis virus (MCoV), human coronavirus (HCoV), feline infectious peritonitis virus (FIP-CoV) and SARS-CoV-1 (Figure 1). The initial results suggest that the SARS-CoV-2 Spike protein shares similarities with SARS-CoV-1, whereas a common ancestor connects MERS and SARS-CoV-2. The majority of similar sequences aligned with SARS-CoV-2 Spike protein are from the Betacoronavirus of the subspecies of Sarbecoviruses, Merbecoviruses, and Embecoviruses; conversely, the minority of them come from Alphacoronaviruses.

Similarly, we identified conserved regions of Spike protein through the sequence alignment among the subjected 75 sequences. Starting from S1 N-terminus Domain (NTD), 10 out of 75 subjected sequences match with its first portion at the 1–200 amino acid (aa) range, whereas 24 out of 75 sequences match with the S1-NTD’s last portion at 201–300 aa. A moderate amount of similarity was observed at the S1 C-terminus Domain (CTD) side with 33 similar sequences at around 300–510 (Figure 2). Moreover, the highest conservation among subjected sequences was observed in the center of the S2 portion of the protein at around 900–1000 aa in 42 out of 75. To strengthen these results, we aligned five Spike proteins among several species related to coronavirus and investigate their conservation in five core domains (S1-NTD, S1-CTD, PCS, S2-HR1, and S2-HR2). The five additional coronavirus species sequences that were subjected to detailed alignment were chosen by their displayed values in the phylogenic tree. As a reference value, we chose the spike protein of novel coronavirus (PDB ID: 6VXX), then we chose three species possessing the highest values and two species possessing the lowest values. This will help us to include a spectrum of proteins that relates to an ancestor protein. The results suggest that the domains of S1-NTD, HR1, and HR2 are highly conserved (Figure 2).

Furthermore, we could identify several conserved regions spanning throughout the Spike protein among the selected viral strains. There are several “clusters” of conserved regions spanning from the beginning S1-CTD until its end at the 330 to 529 aa range. This means that S1-CTD can be divided into seven clusters of conserved domains, composed of around 25 amino acids each, according to the MEGA-X software alignment (Appendix A). This expands our understanding of nature of the Spike protein Binding Domain to ACE2 and other receptors. Moreover, two very large evolutionarily conserved domains lie at the center of the S2 region within the amino acid ranges of 908 to 1003 and 1163 to 1211, indicating that the S2 part of Spike protein seems to be heavily conserved in comparison to its S1 counterpart. Taken together, the S1-CTD and the final portion of the S2 protein are relatively conserved throughout the species and both serve as effective references for the universal drug discovery against most of the strains of SARS-CoV-2 (Figure 2).

### 3.2. Mechanism of Hypoxia Caused by SARS-CoV-2 Spike Protein

We docked the SARS-CoV-2 Spike protein with both the molecule of hemoglobin and the heme group in the ClusPro (v2.0) server and AutoDockTools (v1.5.6) software, respectively. Taking into consideration the docking simulation of hemoglobin, both the monomer (Figure 3) and the tetramer (Appendix A) of hemoglobin bind very tightly to S1-NTD, with several hydrogen bonds bridging the two molecules. The residues of the Spike protein forming hydrogen bonds with the hemoglobin monomer are ARG-237, LEU-270, ASN-87, ARG-88, ASP-627, HIS-625, ASN-61, THR-63, GLN-271, GLN-23, and TYR-28 (Figure 3). Similarly, we found that ARG-237, ARG-21, VAL-53, VAL-6, PHE-4, VAL-3, ASP-111, SER12, GLN-14, ARG-158, CYS-15, THR-22, THR-73, THR-74, GLU-132, LYS-113, and SER-112 of the Spike protein form a hydrogen bond with the tetramer of hemoglobin (Appendix A).

Interestingly, the ARG-237 hydrogen bond persists in both simulations (monomer and tetramer), thus indicating a possible binding area that is more favorable for the interaction in terms of binding efficiency and spatial pairing. Moreover, the region surrounding the above-mentioned amino acid might facilitate the binding of the heme group to the S1-NTD region of the Spike protein. Simulation of this interaction yielded a binding energy score of −5.4 kcal/Mol. The results of the interaction show several Van der Waals bonds surrounding the heme group, whereas there are six Pi-(Anion, Sigma, -Pi Stacked and Alkyl) bonds and one hydrogen bond bridging with the Spike protein (Figure 4A). Amino acid residues TYR-38, LYS-41, PHE-43, LYS-206, GLU-224, PRO-225, LEU-226, VAL-227, ASP-228, and THR-284 of Spike protein interact with the heme group directly. The core amino acid of this interaction is GLU-224, which might possess the ability to directly bind the heme iron with a Pi-Anion bond, thus preventing free oxygen-binding in this area (Figure 4B).

### 3.3. Most Important Conservative Motifs of SARS-CoV-2 Spike Protein as a Target for the Development of Therapeutic Drugs

Most of the antiviral drugs interact with Spike S1-NTD, especially between the residues 1–100 and 201–300 aa (Figure 5A) and with most common interacting residue locations at VAL-3, PHE-4, VAL-6, PHE-58, and PRO-82. On the other hand, antiviral interaction occurs with Spike S2, lying between the residues 801–1000 and 1201–1300 aa. In particular, the amino acids ASN-824, VAL-826, THR-827, LEU-945, LYS-1205, TYR-1209, PRO-1213, and TRP-1217 are very important due to their participance in most of the interactions (Appendix A).

In continuation, interface residues from antibiotic interactions also lie on the S1-NTD between 1–100 and 201–300 aa, which is similar to the antivirals (Figure 5B; Appendix A). Specifically, great importance is possessed by the residues VAL-3, VAL-6, LEU-7, LEU-8, ARG-21, GLN-23, LEU-24, PRO-26, PRO-82, VAL-83, and VAL-289. We also found the interaction pocket at S2, especially residues between 701–800, 901–1000, and 1201–1300, such as GLY-744, ASP-745, LEU-966, VAL-976, LEU-977, ASN-978, and ARG-1000, which play a central role in the interaction with antibiotics (Appendix A). In addition, residues 601–700 at the junction of S1 and S2 seem crucial for the action of antibiotics.

Furthermore, in the case of antiparasitic drugs, we found that the residues of S2 Spike between 901–1000 and 1101–1200 aa are very important, especially the locations LYS-1205 and TYR-1209, which are similar to antivirals and antibiotics (Figure 5C). However, we also found interaction at S1-NTD between the residues 1–101 and 200 aa, which suggests the therapeutic importance related to S1 Spike protein. Additionally, a study with flavonoids revealed that interactions mainly happen at S1-NTD between the residues 1–100, especially at the locations VAL-6 and ARG-21, and at S2 between the residues 701–900, such as LYS-790, LYS-811, ALA-893 and LEU-1224 (Figure 5D). Similarly, the interaction with vitamins C and D lies between the residues 1–100, 301–400, and 501–600 amino acids of Spike S1, most importantly VAL-47, PHE-318, and THR-630, and between the 701–1000 residues of Spike S2, especially the residues TYR-741, ILE-742, CYS-743, and GLY-744 (Figure 5E; Appendix A).

### 3.4. Most Important Motifs of RdRp Protein as a Target for Therapeutic Drugs

Most of the drugs targeting RdRp mainly interact between the amino acid residues 101–200 and 701–800, especially THR-51, ASN-122, PHE-133, LYS-182, CYS-760, THR-761, ALA-771, GLU-780 (Figure 6A,B). Additionally, we also found a subtle interaction between 1–100 and 801–900 amino acid residues (Figure 6B; Appendix A).

## 4. Discussion

In the search for unique conservative regions in the SARS-CoV-2 Spike protein, we performed an alignment of the amino acid sequence of the Spike protein with the BLASTP database (Figure 1; Appendix A). Spike S1 contains a receptor binding domain (RBD) (AA: 449–510) actively binding with ACE2 receptors (Appendix A), which contains several residues evolutionary conserved and dedicated for the interaction between ACE2 and SARS-CoV strains including SARS-Cov-2 [25,26,27]. In the current study, alignment showed that conservation exists in higher levels at S2-HR1 and HR2, moderate conservation at S1-NTD, and PCS, whereas at S1-CTD there are many hypervariable regions (Figure 2). This might suggest that drugs which tend to bind to those regions are more effective in addressing a bigger spectrum of coronaviruses. Interestingly, most of the antiviral top hit drugs, such as Indinavir (ΔG = −9.8 kcal/Mol) and Nelfinavir (ΔG = −9 kcal/Mol), bind near the center of S2 (Figure 2; Appendix A). A similar pattern was observed with antiparasitic drugs Ivermectin B1a/b (ΔG = −9.16/−8.86 kcal/Mol) and vitamin D (ΔG = −5.52 kcal/Mol), whereas, on the other hand, top hit antibiotic drugs did not show preferences at the S2 binding area of the Spike protein (Appendix A). From this perspective, the above-mentioned drugs could possess the ability to physically inhibit the assembly of the Spike protein as well as the viral entry in a pan-CoV targeted manner.

Shuai and colleagues showed that targeting the HR1 domain of Human-CoV Spike protein, which is in the area of S2, showed a significantly reduced viral entry in mouse models [28]. Our docked drugs are also bind in the same region of Spike, supporting the results of earlier reports [28]. Targeting this specific domain with a small molecule could be a pan-CoV target, as its conservation is high among coronavirus species. Turning the attention to conserved areas of Spike protein and not to the hypervariable S1-CTD will lead to a drug that not only has great inhibitory potential but also will possess a broad target activity against multiple coronavirus infection.

Furthermore, taking into consideration another spotlight region of the Spike protein, the polybasic furin cleavage site (PCS), which is located in the boundaries of S1 and S2, was under observation [1]. Throughout our docked library, only one molecule could bind to that area. Vancomycin, which exhibited the lowest binding energy (∆G = −10.2 kcal/Mol) from all our docked drug categories, is the only drug that binds within the region of PCS in an approximate position of amino acid range between 654 and 692. The functional activity is still unknown for SARS-CoV-2 PCS; pathogenicity and transmissibility need to be elucidated as well. Emerging research referred to PCS as an enhancer of cell–cell fusion, whereas cleavage of it results in intra-species infection, with a great example being MERS-CoV [29]. Thus, it is very important to evaluate drugs that can bind to that area of the Spike protein since it might result in desirable outcomes supporting the fight against SARS-CoV-2.

Recently, many anecdotal clinical trial reports have suggested that patients affected by COVID-19 not only have impaired respiratory function but also low oxygen count, causing respiratory distress [7,15,16]. This phenomenon led us to test for a possible binding event between the SARS-CoV-2 Spike protein and the heme group of the hemoglobin molecule. That binding might greatly reduce the oxygen affinity to hemoglobin and increase its degradation rate, as many COVID-19 survivors showed a much lower hemoglobin count compared with those not infected [17,18]. It was also revealed that alternatives currently being considered for COVID-19 treatment by increasing hemoglobin production and increasing hemoglobin availability for oxygen binding and by causing hyperventilation with associated increasing levels of oxygen and decreasing levels of carbon dioxide in the blood may significantly ameliorate COVID-19 respiratory symptoms [30,31].

In line with our findings, Spike protein has the potential to interact with the tetramer and monomer of hemoglobin (Figure 3; Appendix A). Here, amino acid GLU224 might be the key target as it plays a crucial role in binding directly with the iron particle of heme group (Appendix A). This is the first study which shows that the Spike protein of SARS-CoV-2 might interact with not only the hemoglobin molecule but with the heme group as well. These results will shed light on the, yet unknown, intrinsic mechanism of SARS-CoV-2 Spike protein and the low oxygen count of patients and survivors, while, at the same time, while serving as will serve as a basis for the development of new drugs.

Based on the simulated interaction of SARS-CoV-2 Spike protein with crucial antiviral, antibacterial, antiparasitic, flavonoids and vitamins, we found a few striking readouts. Initially, we focused on 47 different antivirals already approved and broadly available in the clinical practice suggesting these have the potential of a viral antagonistic effect. Given the fact that the Spike protein is evolutionarily conserved, it is a prime location for interaction and inhibition of the virus in a manner that is ubiquitous across all potential strains. Our results show a wide range of effectiveness of antivirals drugs, from Saquinavir at the lowest (ΔG = −0.23 kcal/Mol) to Indinavir at the highest (ΔG = −9.8 kcal/Mol) (Appendix A). Drugs with interaction energy greater than or equal to the Japan-made Favipiravir (ΔG = −5.2 kcal/Mol) might be prime candidates to be potential treatments, as it has already been observed to have mild inhibiting effects on COVID-19 patients [32]. Further examination of the results revealed that the drug molecules interacted with the Spike protein monomer at very specific locations (Figure 5A). Most notable are the β-sheets within the 1–100 residues range having over 150 different molecular interactions. This is followed by the α-helix within the 801–900 aa residues range yielding over 100 unique interactions. Noting the location of the 1–100 aa range, drugs that interact with that binding pocket will most likely not have a large-scale effect on the protein’s functionality, as it only serves as the N-terminal domain. Conversely, the 801–900 aa region is located at the point on the Spike monomer that is key to the formation of the Spike trimer; binding to this region would impair SARS-CoV-2’s ability to form the trimeric structure and overall inhibit the production of viable virions [11,12].

Researchers have shown that the combined use of azithromycin and hydroxychloroquine suppressed the growth of SARS-Cov-2 virus in vitro [33,34]. Therefore, similar to the examination of antiviral drugs, we investigated the binding properties of 21 prominent antibiotics to the SARS-CoV-2 Spike protein. Unfortunately, the majority of the molecular interactions observed occur in the N-terminal domain of the Spike protein (Figure 5B), which indicates that they might yield little to no result in disrupting the virus. In conclusion, antibiotic treatment may not play an important role in combinational therapies.

The prospective anti-parasitic, Ivermectin and hydroxychloroquine, the two FDA-approved drugs that were previously shown to have broad-spectrum antiviral activity [33,35]. Leon Caley et al. showed that a single addition of Ivermectin to Vero-hSLAM cells 2 h post-infection with SARS-CoV-2 significantly reduced viral RNA [36]. Ivermectin, therefore, warrants further investigation for possible benefits in humans. Similarly, in our study, we found that Ivermectin B1a and B1b have low energy interactions (ΔG = −9.16 kcal/Mol and ΔG = −8.86 kcal/Mol respectively). Additionally, these interactions were observed to occur primarily on the α-helices existing in the 901–1200 aa region of the Spike monomer (Figure 5C), given the fact that this region is involved in the formation of the Spike trimer, specifically the base of Spike where protein is integrated into the virion, which is conserved among coronavirus species [2,11,12]. Similarly, the 4-aminoquinoline antimalarials chloroquine and hydroxychloroquine have been investigated (Gautret, Lagier et al., 2020), which further support the interaction we found between hydroxychloroquine and the SARS-CoV-2 Spike protein (ΔG = −2.85 kcal/Mol) (Appendix A).

In this line, flavonoids were shown to play a crucial role against different emerging viruses. Tetramethoxyflavone from elderberry extract inhibited human influenza A (H1N1) infection in vitro according to a study of Roschek and Fink [37]. Moreover, flavonoids were utilized against the infectious bronchitis virus (IBV), a pathogenic chicken coronavirus [38]. Chen and colleagues tested non-cytotoxic, crude ethanol extracts of *Sambucus nigra* fruit for anti-IBV activity and showed inhibition of the virus by compromising their envelopes. They also suggest that future studies using *S. nigra* extract to treat or prevent IBV or other coronaviruses are warranted [39]. It has been suggested that flavonoids such as herbacetin, rhoifolin, and pectolinarin were found to effectively block the enzymatic activity of SARS [40,41]. In our study, we focused on five different flavonoids (Appendix A) that were proven to have some effect with other coronaviruses. Although flavonoids did not exhibit high ∆G values, they showed unique binding areas. From the binding areas, we could roughly identify the preferred binding regions on the Spike protein (Figure 5). Total interactions per segment results suggest a tendency of flavonoids to bind in both S1-NTD and S2. Even though most of the screened drugs and small molecules bind with the S1-NTD region, flavonoids showed an increased number of interactions in the conserved S2 region too. This suggests that they might possess the ability not only for structurally inhibiting the assembly of the Spike trimer but also for binding into the lumen of the trimeric protein and thus inhibiting the viral fusion with the host cell in a variety of coronaviruses.

Earlier, researchers claimed that vitamin C and vitamin D have a beneficial effect on lower respiratory tract diseases [42,43]. In the light of the evidence from various groups regarding the beneficiary effect of these vitamins, we investigated their interaction with the Spike protein. There are two forms of vitamin D—cholecalciferol (vitamin D3) and ergocalciferol (vitamin D2) [44]. Here, we found that vitamin C (∆G = −2.95 kcal/Mol) and vitamin D3 (∆G = −5.52 kcal/Mol) can directly interact with the Spike protein of SARS-CoV-2 (Appendix A), which further reaffirmed the possibility of their use as a drug for the treatment of COVID-19. Recently, researchers reported an association between reduced levels of vitamin D and mortality cases caused by COVID-19 in various countries. These results suggest that a higher mean level of vitamin D could results in lower mortality rates [45].

Inhibition of the RNA-dependent RNA polymerase (RdRp) employed by RNA viruses during their replication process [46,47] has been an effective treatment strategy. Japanese anti-influenza drug Favipiravir (sold as Avigan) showed promise in the treatment of COVID-19 patients. However, as the pandemic evolved, it became clear that this drug was only effective if administered at the early stages of coronavirus infection [46]. This suggests that the inhibition of RdRp works primarily as a preventative measure rather than a primary form of treatment against RNA-based viruses. Further, our findings suggest the efficacy of Beclabuvir, in particular, to be used against SARS-CoV-2. Not only does it interact significantly with the viral RdRp, but its binding energy to the Spike protein exceeds that of a significant number of the drugs tested in this study. This suggests that designing a drug which targets similar residues would have a stronger inhibitory effect on the virus than that of Favipiravir and provide additional protection in the form of disrupting the Spike protein’s function. Using Favipiravir as a standard at ΔG = −3.09 kcal/Mol for the ability to exhibit a mild clinical success, three RdRp-targeting drugs are immediately identifiable as potentially more successful: Ribavirin (ΔG = −4.20 kcal/Mol), Galidesivir (ΔG = −4.38 kcal/Mol), and Beclabuvir (ΔG = −5.63 kcal/Mol). Additional examination of where these molecular interactions take place revealed that they are primarily concentrated in the 101–200 and 701–800 aa regions (Figure 6B). Similar to the Spike protein, the 701–800 aa regions in particular consist of a series of α-helices that are essential to the RdRp’s functionality; blocking them would, therefore, inhibit the use of RdRp and essentially shut down the virus’s ability to replicate its genome.

Taken together, analysis of the most interacting motifs of Spike and RdRp, along with evolutionary conserved motifs of SARS-CoV-2 Spike, provided advanced understanding of the computer-aided drug and antibody/vaccine designing for the treatment of COVID-19. Our analysis suggests there is a high possibility of mutation in the S1 region of the Spike protein; in contrast, S2 remains more conservative during the evolution. Therefore, the use of a combination of conserved peptide sequences of Spike protein could be a strategy to develop an effective vaccine. In addition, designing the new drugs selectively targeting these conservative motifs might help to find out a potential cure for the COVID-19. However, prior in vitro/in vivo investigation is warranted to ensure the clinical application of the results of the present study. Conclusively, designing inhibitory peptides blocking the receptor-binding domain of Spike protein interacting with hemoglobin/heme that we reported in the current study could help to control the hypoxia condition during the infection of SARS-CoV-2, which might significantly improve the survival of patients suffering from COVID-19.

## Figures and Tables

**Figure 1 antibodies-10-00003-f001:**
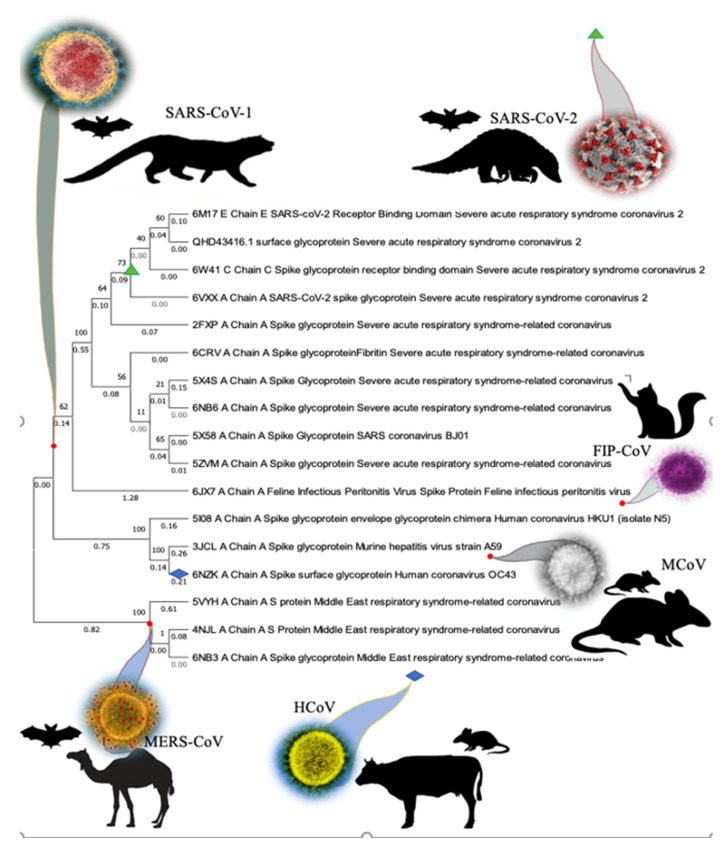
The evolutionary tree was constructed based neighbor-joining and BIONJ algorithms using MEGA 6.0. The topological stability of the NJ tree was achieved by running 100 bootstrapping replications. Bootstrap values (%) are indicated by numbers at the nodes and the length of the branch (represents the evolutionary time between two nodes). This analysis involved 17 amino acid sequences.

**Figure 2 antibodies-10-00003-f002:**
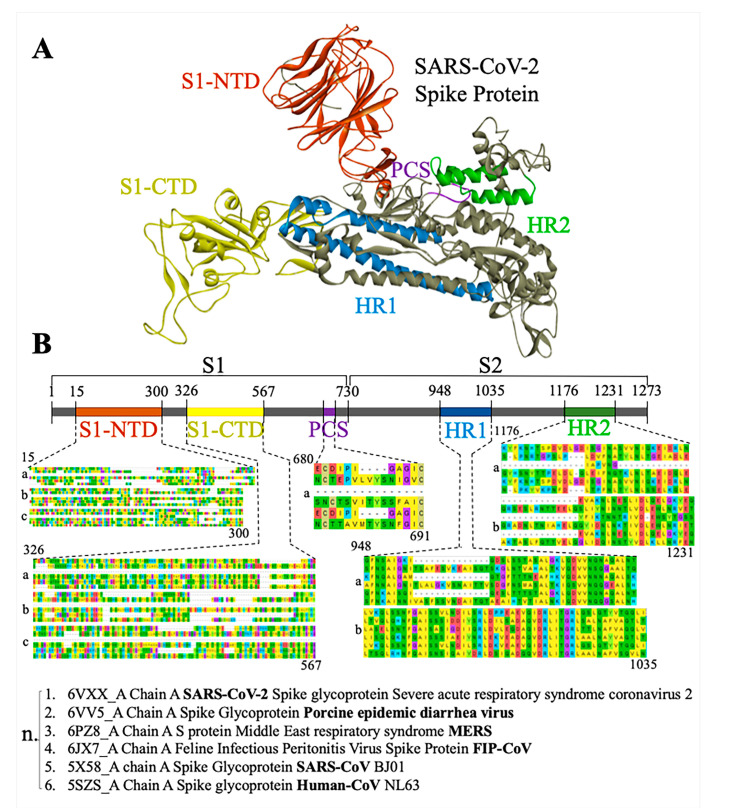
(**A**) SARS-CoV-2 Spike protein 3D structure with the important domains depicted with colors; S1–N Terminal Domain (orange), S1–C Terminal Domain/ ACE2 Binding Domain (yellow), Polybasic Cleavage Site (Purple), S2 Heptad repeat 1 (blue) and S2 heptad repeat 2 (green). (**B**) SARS-CoV-2 Spike protein 2D representation. Important domains are magnified, and their alignment sequences are shown beneath them. Each alignment box consists of 1 to 3 lanes (a, b, c), whereas the starting amino acid is shown at the top left and the ending amino acid is shown at the bottom right of each alignment box. Each lane consists of 6 sequences subjected to the alignment (n). Conservation exists in higher levels at S2-HR1 and HR2, moderate conservation occurs at S1-NTD and PCS, whereas at S1-CTD there are many hyper-variable regions.

**Figure 3 antibodies-10-00003-f003:**
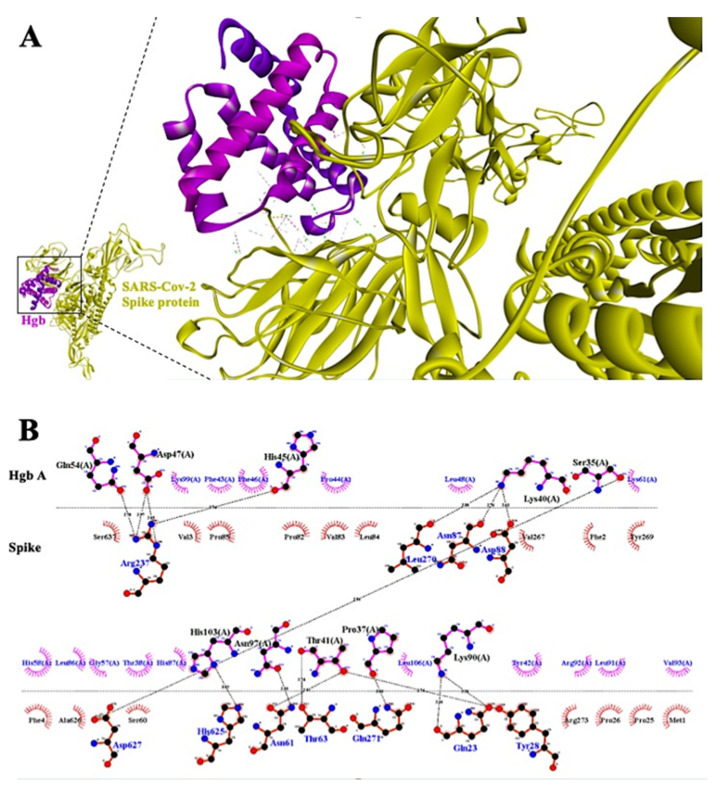
(**A**) Three-dimensional representation of the interaction between the Spike protein (in yellow color) and hemoglobin monomer (in magenta color) and detailed representation of the binding area. (**B**) Two-dimensional interaction plot was formed via LigPlot. Interaction may occur at S1-NTD with the referred amino acids from each side. In the green dashed line, hydrogen bonds are represented and radian arches represent weaker electrostatic bonds that participating in the interaction.

**Figure 4 antibodies-10-00003-f004:**
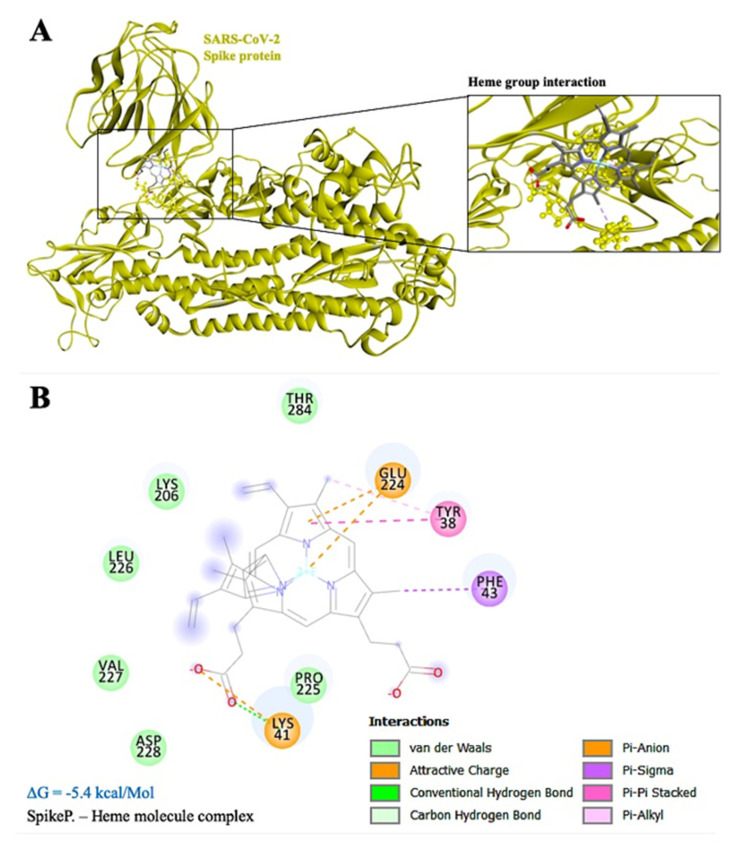
(**A**) Spike protein (in yellow color) docked with hemoglobin heme group near the S1-NTD. (**B**) Two-dimensional interaction plot of the interaction, where GLU224 seems to play a central role in this interaction via direct binding with the iron particle of heme. Binding energy of this reaction was calculated at −5.4 kcal/Mol.

**Figure 5 antibodies-10-00003-f005:**
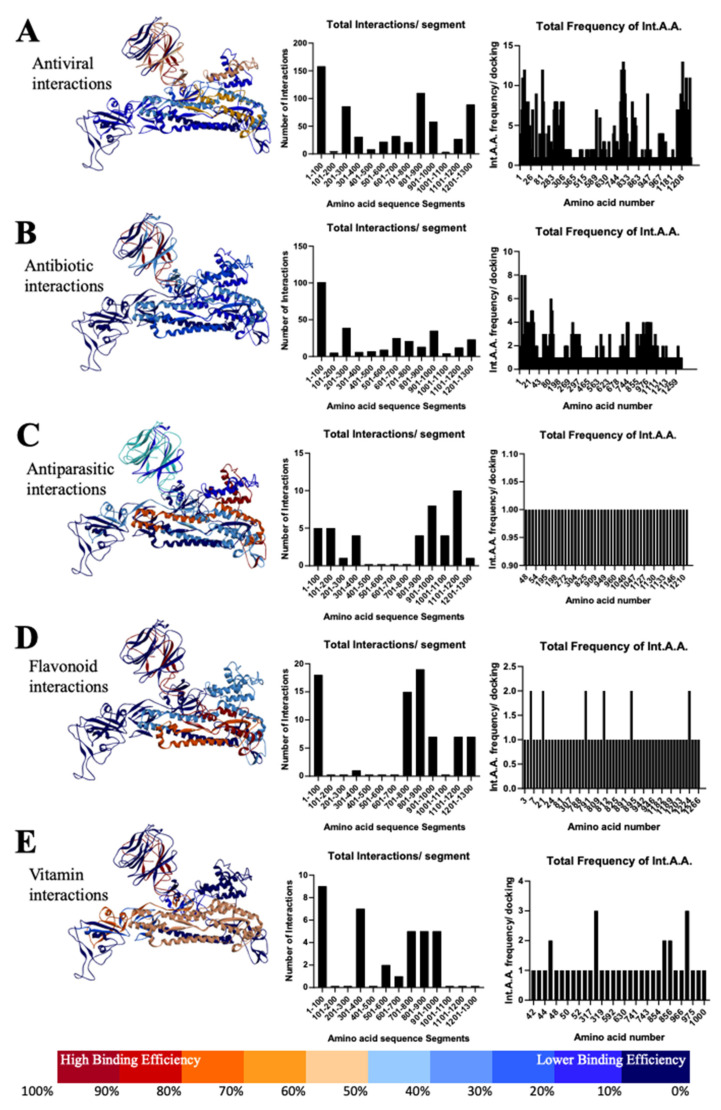
Heat map of the Spike protein represents the frequency of antiviral drug bindings against the 12 segments of the protein. Simulations suggest a preference of antiviral drugs to bind around the S1-NTD region and S2 (**A**); similar results were observed in flavonoid interactions (**D**). Docking of antibiotic drugs exhibits a preference in the S1-NTD region only (**B**). Antiparasitic drugs interestingly showed a higher binding frequency in S2 of the protein (**C**). From the perspective of the vitamins, they were shown to bind in both the S1-NTD/–CTD region and S2 (**E**).

**Figure 6 antibodies-10-00003-f006:**
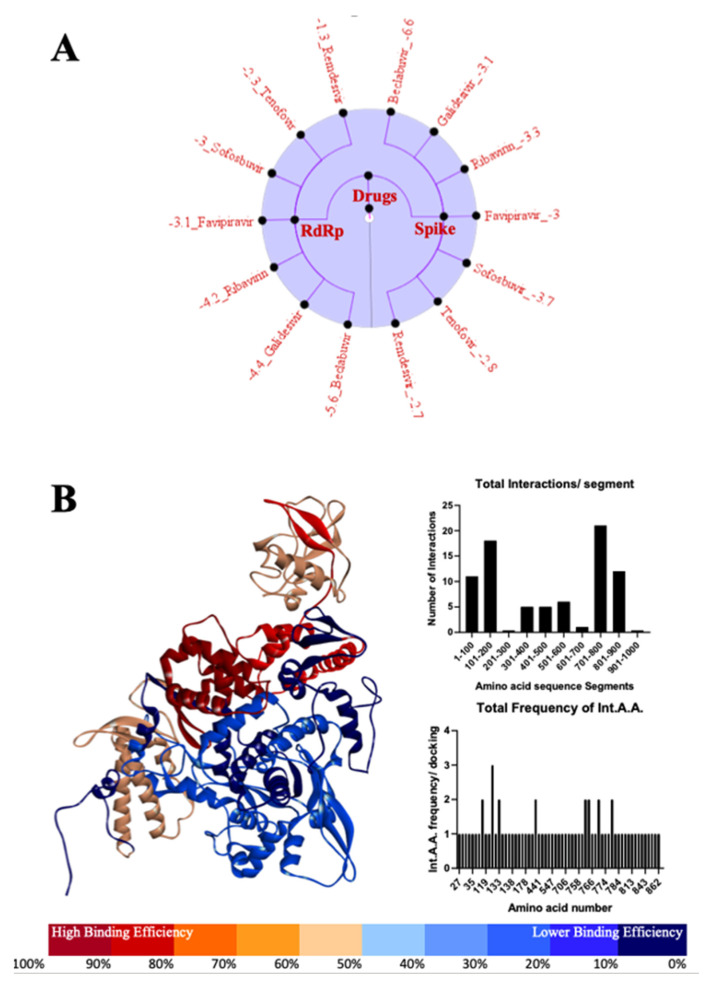
A graphical representation of the comparative analysis found in Appendix A. Binding energies are organized here (**A**) and are arranged from weakest to strongest based on the molecule they are interacting with. These lists of drugs and their binding energies are separated along the central axis based on the viral protein they are interacting with, namely RdRp on the left and the Spike protein on the right-hand side. Additionally, quantitative analysis and graphical representation of the locations at which the molecules are interacting with the SARS-CoV-2 RdRp (**B**) can be observed.

## Data Availability

The authors confirm that the data supporting the findings of this study are available within the article [and/or] its Appendix A. Generated raw data supporting the findings of this study are available from the corresponding author [LA] on request.

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
