# Peer review of "Viroinformatics-Based Analysis of SARS-CoV-2 Core Proteins for Potential Therapeutic Targets"

_2073-4468, 2021, doi:10.3390/antib10010003_

Round 1

Reviewer 1 Report

The manuscript entitled "Viroinformatics-based analysis of SARS-CoV-2 core proteins for potential therapeutic targets" presented computational approaches in assessing potential drug candidates against SARS-COV-2. Though the authors have presented a laudable experimental design, some parts need improvement. 

Here are my comments: 

  1. Line 105-109. How or what is the criteria set by the authors in selecting these 75 subject sequences for analysis? Please add the set of criteria for clarity. 
  2. Figure1/Lines 111 to 113: it is suggested that the authors improve their phylogenetic tree by adding values such as statistical values, bootstrap values that may support the association/likelihood of the branches of the presented tree. 
  3. Line 174-175: Please define ‘similar coronaviruses’ for clarity. Is this the same group representing 75 selected sequences? Please revise for ease of understanding. 
  4. Figure 2/Line 174-184: It is suggested to add in the methodology the strains and source of the sequence, of the representative sequences used for reference.
  5. Line 183-184: The spike alignment shown in figure2b does not represent the conservation generalized by the authors.  
  6. Figure 3A: Suggested to use images of higher resolution, for some of the texts were unreadable. Generally, please improve the figures/legends/choice of color/labels for figures for readers' ease of understanding. Figures should stand alone and be understood readily.
  7. Methodology: the authors need to explain or include how did they select the antivirals included in the study. Also for ethical reasons, cite your sources properly. 
  8. Did the authors found unique conservative regions of evolutionary significance in RDRP? 
  9. Generally: Some parts of the discussion are results. hence it is advised for the authors to reevaluate the manuscript and present their study in their respective sections in a concise manner.

Author Response

Author repose to the #Reviewer-1 comments

The manuscript entitled "Viroinformatics-based analysis of SARS-CoV-2 core proteins for potential therapeutic targets" presented computational approaches in assessing potential drug candidates against SARS-COV-2. Though the authors have presented a laudable experimental design, some parts need improvement. 

Response: We convey our sincere thanks to reviewer #1 for the encouraging review of our work and for supporting the manuscript for publication. The response to the reviewer’s comments is given below:

Here are my comments: 

  1. Line 105-109. How or what is the criteria set by the authors in selecting these 75 subject sequences for analysis? Please add the set of criteria for clarity. 

Response: We appreciate the reviewer’s comment. To make this the point clearer, we revised the METHOD section, please see the changes made at line 106-124 (highlighted in YELLOW). Our selection criteria was based on the highly aligned subject sequences in BLASTP search, which got the highest score.

  1. Figure1/Lines 111 to 113: it is suggested that the authors improve their phylogenetic tree by adding values such as statistical values, bootstrap values that may support the association/likelihood of the branches of the presented tree. 

Response: We appreciate the reviewer’s comment. We have revised the figure 1 to accommodate the suggested changes. The phylogenetic tree was obtained by applying Neighbor-Join and BioNJ algorithms to a matrix of pairwise distances estimated using a JTT model, and then selecting the topology with superior log likelihood with 100 Bootstrap value (lines 113-119, highlighted).

  1. Line 174-175: Please define ‘similar coronaviruses’ for clarity. Is this the same group representing 75 selected sequences? Please revise for ease of understanding. 

Response: We appreciate the reviewer’s comment. Indeed, this is the same group representing 75 selected sequences. To attain greater clarity, we have revised the line 185-186 (highlighted in yellow) to ease of understanding.

  1. Figure 2/Line 174-184: It is suggested to add in the methodology the strains and source of the sequence, of the representative sequences used for reference.

Response: Thank you for the valuable suggestion. As suggested, we added the details of the sequence source and the strains at lines 120-124 (highlighted in yellow) in the methodology sections.

  1. Line 183-184: The spike alignment shown in figure2b does not represent the conservation generalized by the authors. 

Response: We appreciate the reviewer’s comment. To attain clarity of this point, we have amended the manuscript at lines 191-199 (highlighted in yellow).

  1. Figure 3A: Suggested to use images of higher resolution, for some of the texts were unreadable. Generally, please improve the figures/legends/choice of color/labels for figures for readers' ease of understanding. Figures should stand alone and be understood readily.

Response: As suggested, we have revised Figure 3, accordingly to attain greater clarity and details.

  1. Methodology: the authors need to explain or include how did they select the antivirals included in the study. Also, for ethical reasons, cite your sources properly. 

Response: We selected antiviral drugs those readily available in the market and approved by FDA. In addition to that these drugs were under investigation for their potency for the treatment of COVID-19. We selected the drugs from PubChem database (https://pubchem.ncbi.nlm.nih.gov/), which is the national library of medicines managed by National Institute of Health, USA. We amended the manuscripts to incorporate the above details at lines 138-142 (highlighted in yellow).

  1. Did the authors found unique conservative regions of evolutionary significance in RDRP? 

Response: We appreciate the reviewer’s comment. We did not perform the evolutionary analysis for RdRp protein. The main motivation to examine RdRp was the release of data showing that drugs targeting this protein were having moderate success in halting the progression of the novel coronavirus (SARS-CoV-2).  This was included to show the potential for therapeutic targets beyond the Spike protein but not intended to be a primary focus of the study. Rather, it opens the door for future investigations.

  1. Generally: Some parts of the discussion are results. hence it is advised for the authors to reevaluate the manuscript and present their study in their respective sections in a concise manner.

Response: We appreciate the reviewer’s comment. As suggested, we have corrected the discussion section according to refine the outcome of the study (please see lines 314-458, also the highlighted sections).

Reviewer 2 Report

In this present study, the authors concluded that the designing of inhibitory peptides locking the receptor-binding domain of Spike protein interacting with hemoglobin/heme, could help to control the hypoxia condition during the infection of SARS-CoV-2, which might significantly improve the survival of patients suffering from COVID-19. The manuscript is very well written, and it has a nice flow of information. The scientific merits of this manuscript are high. Minor corrections:

In figure 4A, authors can increase the font size of the legends

In Figure 1 the author should arrange the picture in the portrait format

Authors can cite few references that confirm the aminoacids numbers for the Receptor binding Domain of S proteins

Author Response

Author’s repose to the #Reviewer-2 comments

In this present study, the authors concluded that the designing of inhibitory peptides locking the receptor-binding domain of Spike protein interacting with hemoglobin/heme, could help to control the hypoxia condition during the infection of SARS-CoV-2, which might significantly improve the survival of patients suffering from COVID-19. The manuscript is very well written, and it has a nice flow of information. The scientific merits of this manuscript are high.

Response: We convey our sincere thanks to reviewer #2 for his strong and encouraging comments on our submitted work.  We have made the amendments suggested by the reviewer, enlisted as follows:

Minor corrections:

  1. In figure 4A, authors can increase the font size of the legends.

Response: As suggested, we have revised the figure 4A legends. We thank the reviewer for the valuable suggestion.

  1. In Figure 1 the author should arrange the picture in the portrait format.

Response: As suggested by the reviewer, we have revised the figure 1. We thank the reviewer for the valuable suggestion.

  1. Authors can cite few references that confirm the aminoacids numbers for the Receptor binding Domain of S proteins.

Response: We appreciate the reviewer’s suggestion. We have revised the manuscript according to the suggestion. In the present study, we found that Spike S1 contains a receptor-binding domain (RBD) (AA: 449-510) actively bind with ACE2 receptor, contains several residues evolutionary conserved and dedicated for the interaction between ACE2 and other SARS-CoV strains including SARS-CoV-2 (Ali & Vijayan, 2020; Chakraborti, Prabakaran, Xiao, & Dimitrov, 2005; Yang et al., 2020), which further strongly supports the result of the current study. Please see the revised text on lines 316-318.